# Secure Access Control Realization Based on Self-Sovereign Identity for Cloud CDM

Yunhee Kang [1] and Young B. Park [2,*]

1    Division of Computer Engineering, Baekseok University, Cheonan 31065, Korea
2    Department of Software Science, Dankook University, Yongin 16891, Korea
*    Correspondence: ybpark@dankook.ac.kr

**Featured Application: The proposed access control is designed for handling a permission of a researcher based with capability represented as a credential. It provides the identity management including privacy preserving, interoperability, trust in cloud CDM.**

**Abstract:** Public healthcare has transformed from treatment to preventive care and disease management. The Common Data Model (CDM) provides a standard data structure defined to utilize hospitals' data. Digital identity takes a significant role as the body of information about an individual used by computer systems to identify and establish trust among organizations. The CDM research network, composed of users handling medical information, has several digital identities associated with their activity. A high central authority cost can be reduced by Distributed Ledger Technology (DLT). It enables users to control their identities independently of a third party. To preserve the privacy of researchers in clinical studies, secure identification is the main concern of identifying the researcher and its agents. To do so, they should pose a legally verifiable credential in the cloud CDM. By presenting the proof represented by the capability that the user has, each identity has access control that is linked to an authentication credential that the cloud CDM can verify. Assurance in one's identity is confirmed by asserting claims with the identity and its capability, providing its verifiable credential to the authentication entity in the cloud CDM. This paper describes the user-centric claim-based identity operation model based on use cases to handle researcher identity in the cloud CDM. In this model, credentials are designed as a capability and presented to them to access SPs in the cloud CDM. To provide well-controlled access control in the cloud CDM, we build and prototype a capability based CDM management system.

**Keywords:** common data model; digital identity; distributed ledger; identification; claim-based identity

## 1. Introduction

Public healthcare has transformed from treatment to proactive prevention and management of disease. Clinical research is based on patient data derived from hospitals participating in a distributed research network [1]. Observational databases vary in both purpose and design. To enhance the collaborative work in the distributed network, interoperability in healthcare is focused on exchanging healthcare data between multiple hospital systems in a distributed research network. The Common Data Model (CDM) represents healthcare data from diverse health records from hospitals. It is used for observational research in a distributed research network [2,3]. It consists of international standard terms with different elements according to the purpose [2–4]. CDM is a data standard that is proposed to standardize the structure and content of observational data. It is used to enable distributed collaborative research and support to enable efficient analyses that can produce reliable evidence from a variety of data sources across multiple institutions. The Observational Health Data Sciences and Informatics (OHDSI) community proposes OMOP, one of the representative CDM standards, which allows for the systematic analysis of disparate

observational databases. With the increasing volume of electronic data in the field of health-care, provider organizations need to handle identity governance to proactively mitigate high-risk access before a personal data breach occurs. A data breach can be conducted by attacks, including stealing data through insider leaks or exploiting software or hardware vulnerabilities [5]. Distributed storing encrypted data is required to prevent those problems. To overcome the problem caused by data breach, we generate the encrypted CDM and securely deliver it to IPFS (Inter Planetary File System) managed by the cloud CDM. The cloud CDM is introduced to support CDM derived from HIS (Hospital Information System) and its operation scheme among hospitals in the cloud CDM [6].

Digital identity takes a crucial role as the body of information about an individual used by computer systems to identify and establish trust among organizations [7,8]. A distributed research network composed of researchers handling medical information has several digital identities associated with their activity. An identification scheme to preserve the role of a researcher is required.

Blockchain has tamper-proof Distributed Ledger Technology (DLT), which is used to record the transaction permanently [9–11]. Blockchain is originated to regard decentralized control and immutability of event logs [9]. It suggests adopting a distributed peer-to-peer timestamp server to solve the double-spending issue. With regard to the transactions recorded in the ledger, blockchain fosters benefits, including auditability, chain traceability, and high transparency. Its characteristics contain consensus mechanisms, privacy set- tings, transparency, immutability, and decentralized control [12]. OmniPHR focuses on personal health records, which allow people to manage their medical records, distribution, and interoperability. It entails dividing PHR into data chunks that are dispersed over a P2P network [4]. The high cost of a central authority of identity management can be reduced by DLT. Therefore, we consider utilizing the distributed ledger and the node-to-node consensus method to build a trusted digital identity management system in an untrusted environment. The issues of identity fragmentation, theft, and single points of failure in conventional centralized identity management systems are resolved by enabling users to manage their identities independently of a third party [13,14]. Due to the features of the distributed ledger, which enable institutions to communicate with one another while maintaining the requirement that each institution always abides by all applicable legislation, DLT provides an alternate solution.

Recently the European General Data Protection Regulation (GDPR) allowed individ-uals the Right To Be Forgotten(RTBF), meaning once the reason for storage has passed, they can request that their personal data be erased at any time [15–17]. The developing blockchain technology has made it possible to find solutions to issues related to the man-agement of identity; however, storing personal information on the blockchain may not be a solution [15].

To preserve the privacy of researchers in clinical studies, secure identification is the main concern of identifying the researcher and its agents. To do so, they should pose a legally verifiable credential in the cloud CDM. By presenting the access capability that the user has, which is associated with a credential that can be verified by the cloud CDM, each identity acquires access control. Assurance in an identity is established by asserting claims with the identity and its capability, providing its credential to the authentication entity in the cloud CDM. Personal information of each researcher is decentralized through autonomous control of his/her digital identity.

This paper describes the user-centric claim-based identity operation model based on given user cases to prove researcher identity in the cloud CDM. In this model, credentials are designed as a capability and presented to access services of different institutes in the cloud CDM. To provide well-controlled access control in the cloud CDM, we build and prototype a capability based CDM management system. In the case of multiple medical organization data access control, a credential for each researcher is defined, enabling access to services related to CDM.

## 1.1. Problem Statements

Information systems are required to be secure in the presence of wrong accesses of subjects when they operate transactions to objects such as files. Registering and maintaining the identity and qualification of researchers from various medical institutions through the research portal to access CDM data has difficulties such as the risk of exposing personal information of researchers and managing CDM data by level. In the cloud CDM, researchers are issued credentials with attributes of access rights called capability. The authenticated subject with permission can manipulate CDM data of the cloud CDM only in the authorized operations. Capability is a collection of access rights on CDM in a cloud CDM. When researchers perform operations such as querying, manipulating, and accessing health records in the cloud CDM, capability is attached to indicate that the operations are authorized. Traditional models cannot block the unnecessary flow of private information in the perspective of identity management. We propose a model that blocks transactions where improper information flow occurs in the cloud CDM. Each agency in the cloud CDM must define the role of a given subject in the process and check it before executing each task. The concept of user-centric identity management gives a way to establish a comprehensive and self-described control over access that users have over all of their data and certifications. It is analogous to how identity is managed today by using a system of physical wallets and plastic cards. The purpose of the digital identity management system is to achieve a control of workflow to handle CDM data without a distinct central authority. An essential component of an access control procedure for managing CDM data is safeguarding researchers' privacy.

In the use case of utilizing CDM data, the Institutional Review Board (IRB) must approve requests for CDM access permission regarding researchers before the data supervision process can proceed. The process's outcomes include working on the access control of a service when various hospitals and healthcare institutes participate in a distributed research network. The compatibility of the collaborating institution's systems and the researcher's authorization to engage in the research must both be demonstrated by the access control system.

## 1.2. Contributions

The following is a summary of our work's main contributions:

1. The traditional identity systems have security vulnerability and depend on trust in third-party authority. Those systems may cause a lack of autonomy, single-point failure, and privacy leaks when we apply them to the cloud CDM. Self-sovereign identity (SSI) allows researchers to provide their identity information with the confidential token controlled by themselves. To design the cloud CDM, we apply SSI as an identity management model that offers a way to handle the programs caused by a third-party authority.

2. The main purpose of the cloud CDM is to collaborate with hospitals and research institutes in order to share Electronic Medical Records (EMR) with diverse formats. EMRs contain confidential private information about patients. Secure and reliable identity management is one of the basic requirements for building an e-health service in the cloud CDM participating in multiple institutes. To build the cloud CDM, the enhanced decentralized identifier and verifiable credential approach prevents disclosing who owns the privacy attribute in the user's wallet while defining it. This proposed model acknowledges the de-linkability between the users and their identity.

3. In this proposed model, a user must demonstrate ownership of the identifier in order to access the service provided by the service provider. This is often done by making a transfer from the wallet associated with the user's blockchain account. The service provider can then retrieve the entire identification.

## 2. Identity Model

Identity management is used for assuring that individuals receive the appropriate access to resources. It handles the overall methods and regulations that handle the lifecycle of identity traits for a specific domain [7,18]. Attributes required by authorizing services that describe individuals and use of that data can be diverse. An ethical committee such as IRB can require researcher's personal attributes and a research plan before it can process a request and permit use of sensitive data for a particular research purpose. To choose the right type of system from the point of view of a cloud CDM, we summarize identity models and their systems in research institutes and hospitals.

Many of the use cases identified by the research communities call for personal information to be aggregated with community defined attributes in order to grant access to digital resources and services. Compared with existing use cases, this aggregation means that an external attribute authority that is typically managed by the research community itself is needed in addition to the IdP and SP. Based on the level of authorization within the different services depends on the credentials presented by the user in the cloud CDM. FIM has a limitation to support assurance and enforcement in terms of the level of security within systems and across boundaries from one to another.

In a centralized or federated identity model, users create accounts to access systems and authenticate transactions with many different identity providers. Organizations collect and store personal and sensitive information about their users to enable services. This results in our digital identities being spread all over the web, making ID theft, credential compromise, and breaches a real possibility with potentially significant long-term impacts for both individuals and businesses. However, it is not easy to protect the problem caused by security breaches resulting in unauthorized access to personal data in the organization.

The distributed ID type is operated in such a way that users create their own identification information and share it through DLT. The convenience of identity management can be improved as the procedure for registering verification information becomes unnecessary. In terms of the safety of the security system, the user's identity is not stored in the central server, so there is little risk of security leakage or theft, and it has the advantage of ensuring the integrity of the identification information. Self-sovereign identity refers to a blockchain-based distributed self-sovereign identity management technology that allows users to directly manage their identity and to select the subject and the scope of disclosure. Distributed ID is used as a technique for this. Self-sovereign identity systems are decentralized and operate in the same way as real-life identities because they do not depend on a central authority. The user has all the rights such as issuance, submission, deletion, and recovery of the identity certificate, and the blockchain proves that the identity certificate is owned by the user and is used for record management such as issuance, submission, and renewal of the identity certificate.

The baseline of identity management architecture is shown in Figure 1. It is made up of the user or object, a service provider (SP), and an identity provider (IdP). Identity management systems are used by the SPs to authorize users and to give them access to the services.

- A subject must submit a request for an identity from the identity manager in order to access the desired service. Based on the data the subject has provided, the identity manager then creates a distinct identity and responds to the subject.
- The subject demands a certain service from the service provider, who then asks the subject for identification details. The request is received by the subject, who then responds with the pertinent information.
- The service provider asks the identity provider to confirm that the user's identification is legitimate. The service provider offers the service based on the received validation results after the identity provider returns the authentication results.

1. IdP: persists user identity, authenticates and authorizes incoming requests.
2. SP: Has resources. Access to the resources is restricted by IdP.
3. Subject: Has a credential used for authentication. Subjects serve as the system's main facilitators, utilizing the numerous services provided by the service provider and identity provider.

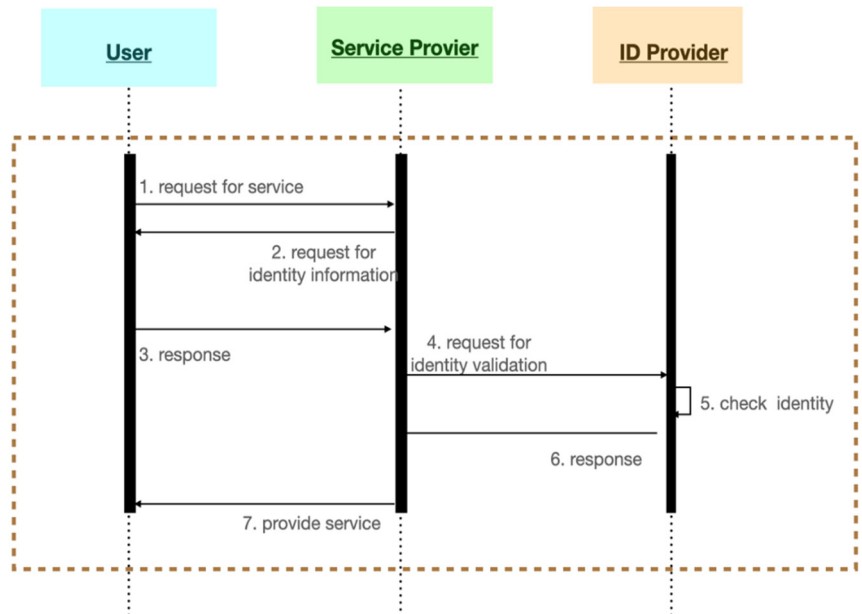

**Figure 1.** Traditional identity management architecture.

Type of Identity Management (IdM) can be classified into three models: Isolated IdM, Centralized IdM, and Federated IdM [7]. Table 1 shows the types of IdM.

**Table 1.** Characteristics of IdM.

|  | Isolated IdM | Centralized IdM | Federated IdM |
|---|---|---|---|
| Identity Management Autdority | Service provider | Centralized identity management server | User-selected identity provider |
| Registration of Identity information | User-attribute registration by service | Central identity server | Pre-registration witd IdP as federation service user |
| Privacy preserving | Privacy-leak problem | Privacy-leak problem | Privacy-leak problem |
| Fault tolerance | Single-point failure | Single-point failure | Single-point failure |
| Level of scalability | Low | Low | Middle |

In the Isolated IdM model, also called the Silo model, each service provider has its own IdP. The identity provider and service provider are mixed up and they share the same space. The end user should log in for each service provider with a different identity. The intricacy and inadequate scalability innate to the isolated Id management model leads to relying on the third-party authentication scheme [10]. The identity is handled for each service, and the user needs to register his/her identity with personal information and then to perform a specific authentication process for each service. For user authentication and access control of the cloud CDM service, an identity management system is established and operated with high management cost, because the users of the cloud CDM must perform identity registration separate identifiers for each service.

In the centralized IdM Model, IdP is based on centralized digital identity management. The end user can log in and access with a single user identity. The user registry is managed by third-party services that contain identity information for participants in the cloud

CDM. All of the access controls in all the SPs are controlled by a single access control list (ACL); hence, the users' identities and passwords can be used to authenticate themselves with SPs. The centralized model is weak as it runs the risk of having one identity and having its related credential disclosed. The centralized IdM Model is used to handle access control in one environment. A centralized identity management system empowers teams with visibility so they can detect and respond to threats swiftly and efficiently. It centrally manages reliable user identities and, compared with the isolated type, the establishment and operation of the identity management system is more efficient. Several services connected by authentication are available. With these technical characteristics, the centralized type is suitable for providing multiple services through a single cloud CDM. However, when the central management system fails, it is impossible to use the entire service and there are limitations in interoperability and scalability due to the closed structure. The service provider has to decide whether the cloud CDM should be usable in an inter-institute context in the cloud service. From the point of view of a service provider, it is important to support the ease of deployment of the identity system, preserving privacy.

From an end user's perspective, whether it is a federated IdM model with a specialized trusted IdP in charge of gathering and providing users' identity information or it is a centralized IdM model, it is the same [14]. The end user logs into the service provider with a single user identity. Often found in a secure domain, this model supports Single Sign On (SSO) and distributes user-identification data to various SPs. However, the IdP faces the issue of a single point of failure if the entire identity management system fails like the centralized IdM model. However, each service provider has a different IdP in the backend. To support SSO to users joined to distinct IdPs and SPs, it is necessary to create a trust relationship between the SPs and the IdPs. Authentication is integrated by SSO and authorization is covered by entitlement management with XACML [18–20]. However, SSO also has a problem with the reliance on the IdP portal and SP in terms of access control of the cloud CDM. The federated IdM model is a method in which different service providers form a trust relationship for user convenience and jointly manage the user's identity. It is suitable for SSO support between hospital and cloud CDM networks. However, for federated identity management, it is necessary to establish a trust relationship between hospitals and cloud CDM, and when identity management is concentrated on a specific service provider, it has the same limitations as the centralized type. Federated IdM systems allow users at separate institutes to use the same verification method for access to applications and other resources. Identity federation delivers efficiency to institutes, as it allows them to streamline the processes for verifying their users. This federated IdM model implies a separation of the authentication performed by the IdP and the authorization performed on behalf of the SP. In the federated identity model, some identity federations require the IdP administrators to explicitly give permission to pass on the user attributes to any SP.

In a centralized or federated identity model, users create accounts to access systems and to authenticate transactions with many different identity providers. Organizations collect and store personal and sensitive information about their users to enable services. This results in our digital identities being spread all over the web, making identity theft, credential compromise, and breaches a real possibility with potentially significant long-term impacts for both individuals and businesses. However, it is not easy to protect the problem caused by security breaches resulting in unauthorized access to personal data in the organization.

In the cloud CDM, most SPs play a role as IdPs, as shown in Figure 2. If they desire to use services from the SP, such dual personality compels users to register within its domain [18]. For a better understanding, the following is the operation scheme in a cloud CDM. A researcher has the researcher ID of the researcher portal and submits the researcher information to the IRB. The IRB stores the researcher information in the research workflow based on the researcher information. After that, the IRB provides the researcher with approval or refusal to conduct the research. The investigator uses their investigator ID

to access the CDM provider, which passes the investigator ID to the IRB. After the IRB confirms the researcher's eligibility to conduct the research through the research workflow, the CDM provider sends approval or refusal to the CDM provider. The CDM provider, which plays the role of an SP in the corresponding cloud CDM, depends on the research workflow of the IRB to access its own service. The current central identity management method delegates the management responsibility of the IRB to the researcher, and the delegation of the IRB follows the processing flow of the central model and has difficulties in the detailed management of the researcher's qualifications. It has a security vulnerability by using only the fixed researcher ID for user authentication.

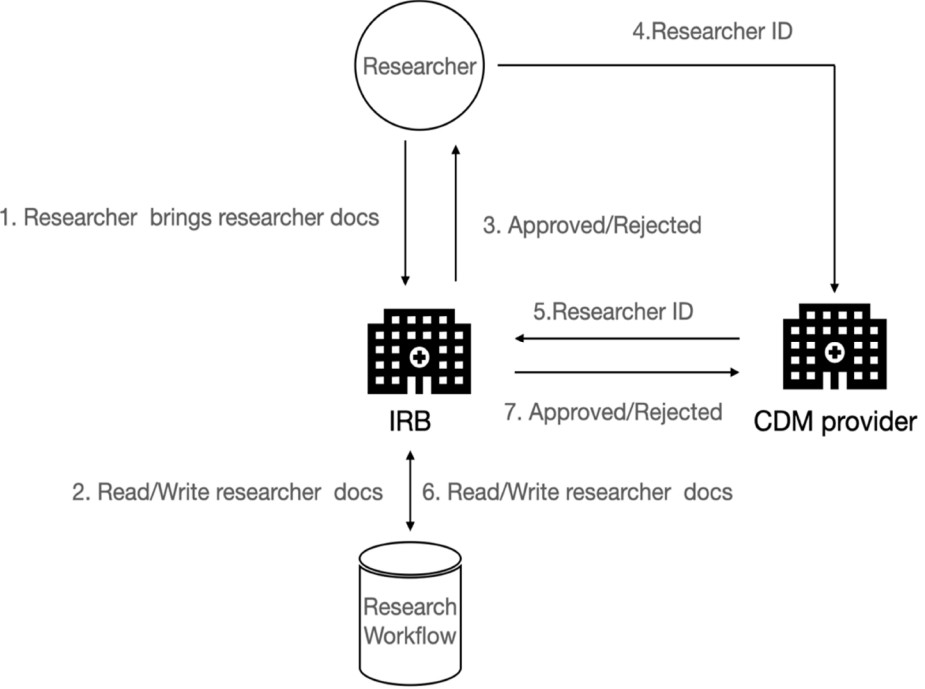

**Figure 2.** IdM based on a centralized model in the cloud CDM.

## 3. The SSI-Based Identity Management

### 3.1. Basic Concept of SSI

According to the World Wide Web Consortium (W3C) working group on verifiable claims, users exist apart from services in a self-sovereign identity (SSI) system. SSI is user-centric in comparison to traditional identity management systems [21]. With the advent of the SSI management system, the introduction of a decentralized management system for identity is being considered. Decentralized Identifiers (DIDs) are established as digital identifiers in a W3C working group that is used to enable trust in distributed environments.

It is used to give an individual rights about his/her private attributes that can be controlled without the bridge via a third party [22]. As SSI requires minimum disclosure in principle, a new format consisting only of information that needs proof is required, which is called a verifiable presentation. By using SSI, claims-based identity using proof eliminates the function for applications to operate authentication tasks, making account management without a centralizing identity management system [23]. In order to achieve the clear mapping of identities representing the same entity with diverse roles maintained in several locations, it is necessary to lower the cost of maintaining identity repositories that need to enforce consistency.

In general, the IdP has the authority to validate the attribute value and to publish claims on the mapping to certify the user's ownership of the property. User identification, locating user identities in detail, keeping track of user passwords and accounts, and interacting with other IdP systems do not need to fall under the purview of SP. However, external users in a cloud CDM organization can access the network applications of another

institute using their own identities thanks to privacy-preserving identity with claims [21,24]. Because companies can construct additional traits as claims upon which to build access control, claim-based identification provides more freedom.

SSI can be interpreted as an identity that has given itself authority. Its system uses blockchain to look up user's identifier without a centralized IdP. When a user registers his/her identifier in the blockchain, the identifier and its attributes are issued and then stored in his/her wallet. Blockchains can offer neutral gateways for cross-organizational digital workflows in the SSI design [25,26], reducing the risk of disclosure of private information. It enables the user to have total control over their unique characteristics. Selective submission of the identity information needed for the service is possible through the channel, and it is also possible to demonstrate the veracity of the given information without involving a third party.

In this paper, we take a look at a repository that enables a ledger where documents can be held fully secretly. This could enable the creation of a blockchain-based decentralized permissioned database. In such a system, the document with private information can only be stored in a personal wallet and managed in a blockchain by the defined processes, including proving and verifying, and not on a central database managed by the regulator. Figure 3 depicts the SSI-based Identity Management in cloud CDM.

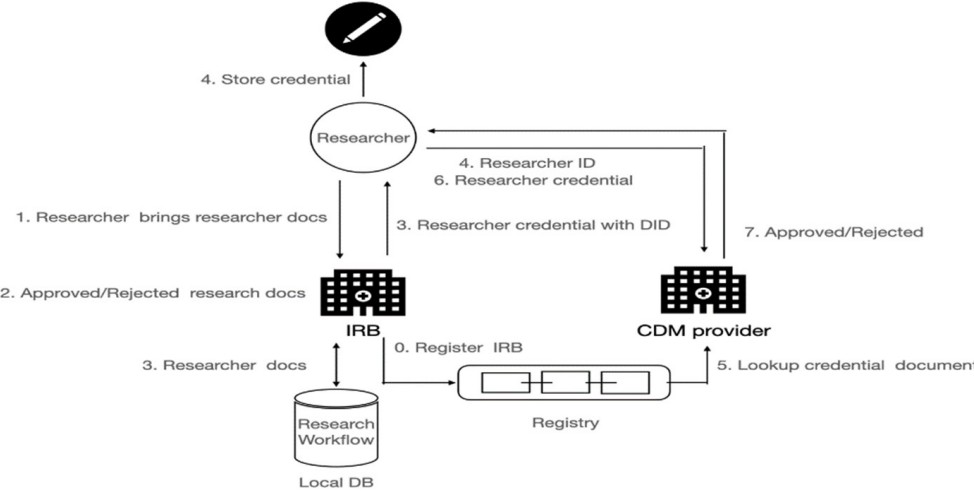

**Figure 3.** The is based on SSI-based Identity Management in a cloud CDM.

### 3.2. Applying SSI Model to Cloud CDM

According to the fundamental SSI model, the personalization agency issues and submits users' identity information in advance so they can acquire the requested service from the verification agency [20]. The certificate of the personalization agent is kept in a verified data registry to guarantee the accuracy of the identification information issued by the personalization agent. When a verification body receives the identification documentation, it authenticates it in the register and offers services. A credential is a group of assertions made up of one or more attributes, such as a name or a resident identification number, that the issuer can verify. It is a declaration of authority, qualification, or competence made to an entity (e.g., an individual or organization) by a third party who has the necessary de jure authority or competence.

An IRB must assess and approve any research involving human subjects or that is subject to FDA regulation. Before beginning any human subject research projects, it is the duty of all researchers to obtain IRB permission or exempt determination in a cloud CDM.

The capability-based trust paradigm for a cloud CDM is depicted in Figure 4. Verifiers are the CDM provider and the CDM customer. With no verifiable credential (VC) granted by IRB, the verifiers may not be able to fully trust the researcher and may act as an entity to release only a portion of the data or to respond to a request for data via the cloud CDM. They might also wish to provide the researcher access to certain data subsets. As

evidence of the researcher's aptitude, the proof assembled from a portion of VC is used. The permission granted by the proof may have to be withdrawn, amended, or put on hold.

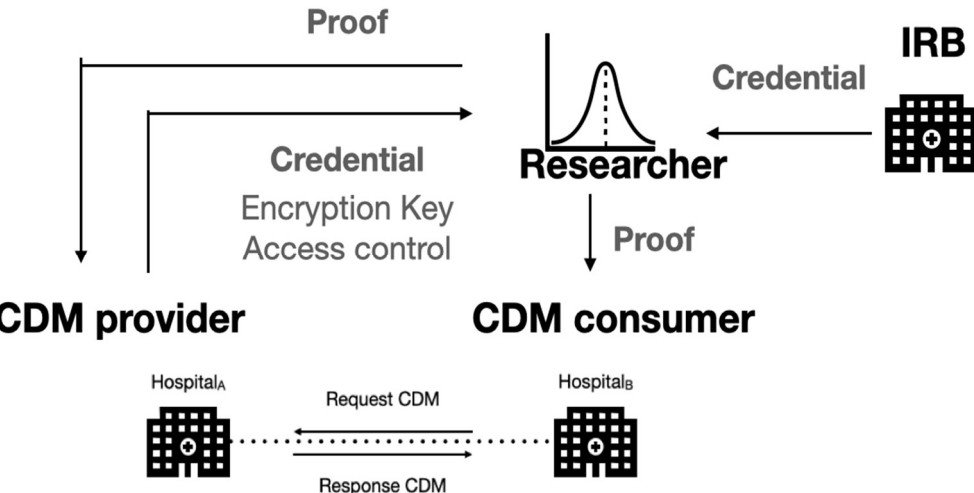

**Figure 4.** The capability-based trust model for a cloud CDM.

Table 2 summarizes the qualitative comparison of the characteristics of the baseline CDM model and the proposed cloud CDM model. In terms of the fault tolerance, data sharing, privacy, and data integrity, it can be seen that the proposed cloud CDM model is improved over the baseline CDM model.

**Table 2.** Qualitative comparison of CDM models.

|  | The Basic CDM Model | The Proposed Cloud CDM Model |
|---|---|---|
| Security | There is no access control mechanism | Supporting high security with access control by verifiable credential |
| Privacy | The exposure of CDM data ownership | Enhanced secure DID ownership |
| Data integrity | Vulnerability of CDM data integrity | Ensuring data integrity with cryptography |
| Data sharing | There is no data-sharing policy | Providing a data-sharing policy through proof and verification by verifiable credential |

There are three entities, IRB, CDM provider, and CDM consumer, in Figure 5. Each of those entities has a public DID that is published globally. The IRB acts as a verifiable credential issuer for a researcher. The identity of the issuer (via the public DID) in the credential ought to be part of what the verifier discovers from the presentation, because the researcher, as the bearer of the credential, may show the credential to anybody. The verifier might investigate to determine whether they believe the issuer. IRB's public DID is placed on a blockchain in order to be resolved globally. It is used to provide secure point-to-point communication channels between the participants' agents. DIDs are used as the identification for IRB as the issuer in the cloud CDM and have a verifiable credential. A public key associated with the DID is obtained by resolving the IRB (the issuer) DID, which is used to specifically identify the issuer. The information in the verified credential is then used with that public key to confirm that it did indeed originate from the issuer. This public DID guarantees that the verifier is aware of the source of the credential that the holder is presenting.

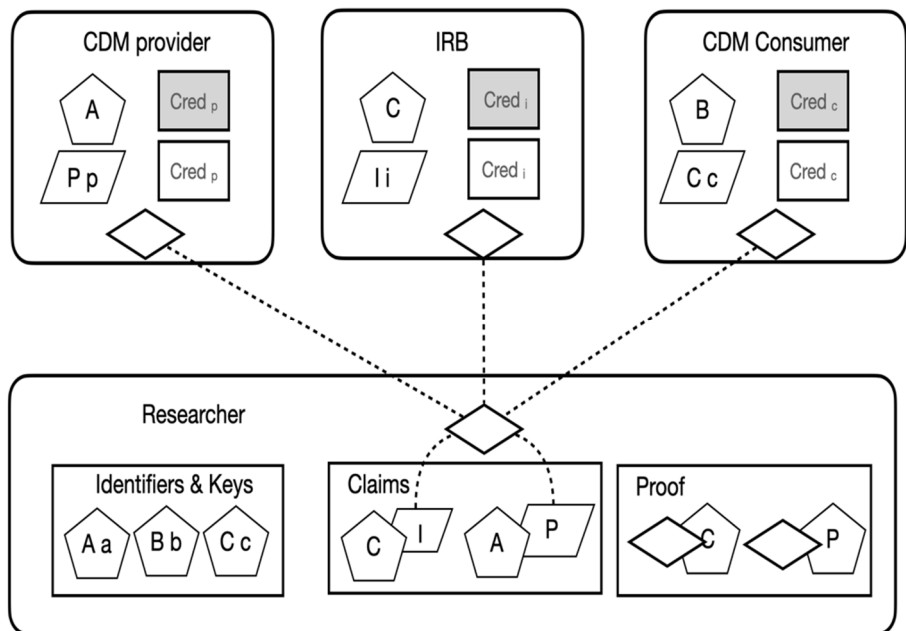

**Figure 5.** The capability based Identity Management in a cloud CDM.

In the cloud CDM, the following application use cases must be used to issue and verify credentials:

- A researcher is a member of a group of researchers who are participating in a specific research project using clinical data. The researcher is given permission through IRB approval and is registered as a permissioned researcher. When the IRB approves the researchers, they are provided with the capability of research participation.
- The researchers can work with given permission when they generate CDM data given from EMR, encrypt the generated CDM data, and obtain new permission with proof for handing the encrypted CDM data in distributed storage.
- The researchers take control of the consumer service of CDM for decryption and distribute the location of the CDM data storage while retrieving previously produced CDM data.

### 3.3. Attribute Definition of Subject

SSI is made up of identifier and identifier's data. In the cloud CDM, identifiers make use of DID, and the identifier data is made up of a variety of attribute data. Personal information, qualifications, and a verified presentation are the three fundamental elements of identity. Figure 6 shows schemas defined by an IRB and a CDM provider. An individual's or an organization's identity as a legal entity can be reflected by a number of attributes (such as name or role in IRB). Various attribute information expresses the identification of the CDM serving and consuming institutions as well as the participants of these institutions. The functions for managing identity data and associated access control are provided by identity management. Basic attributes for obtaining access to IRB services are defined in the schema as shown in Figure 6a. The schema is used for representing the certificate issued by the IRB. Each of the schemas are contained in a transaction of Indy. The CDM provider also defines the attributes for obtaining access to its service written in JSON by a researcher, as shown in Figure 6b.

### 3.4. Issuing Credential

A researcher has their claims verified (and trusted) by medical institutions and a professional accreditation credential from the appropriate organization, such as the College of Physicians and Surgeons. Hence, according to the IRB protocol, IRB issues a credential for a specific service in the cloud CDM. The credential is owned by an individual and is

separated from the issuer, so there is no leakage of personal information or a single point of failure due to the negligence of the certification authority. The capability of the service access is represented with a credential, which is defined and issued by SPs. The main characteristics of a credential are machine verifiability, secureness, and being tamper-proof. To issue a credential, the holder (researcher) sends a proposal to the issuers (IRB and CDM provider). To send a proposal by the holder, the holder uses a protocol for supporting DID communication. After receiving the proposal, the issuer sends an offer to the holder. The issue of credentials is fully automated if the holder automatically takes the offer and converts it into a request.

```
        "IRB_no",                          "approved_date",
        "name",                            "timestamp",
        "role",                            "seq",
        "affiliation",                     "IRB_no",
        "GCP",                             "iv"
        "approved_date"                  ],
      ],                                   "name": "Provider schema",
      "name": "IRB schema",                "version": "29.27.69"
      "version": "56.57.50"             }
    }                                   }
  }
```

| (**a**) | (**b**) |

**Figure 6.** The schema definition written in JSON. (**a**) IRB schema; (**b**) CDM provider schema.

Verifiable credentials are digitally signed by the issuer. A DID registry manages a repository for maintaining DIDs and DID documents. When creating, deleting, or updating a DID or DID document, it is recorded in the data in the DID registry. The DID document describes the contents of the description of the DID subject. This includes data such as public keys or similar biometrics that can be used to authenticate themselves and prove association with the DID. The signature can be validated using the public key. Additionally, the DID document includes the DID identifier and service endpoints for the subject to interact with. The issuer sends a credential to the holder and the holder stores it in their wallet. The issuer should register its DID with its document in the DID registry before issuing the credential to the researcher. Figure 7 shows the process of issuing and verifying a credential in a cloud CDM.

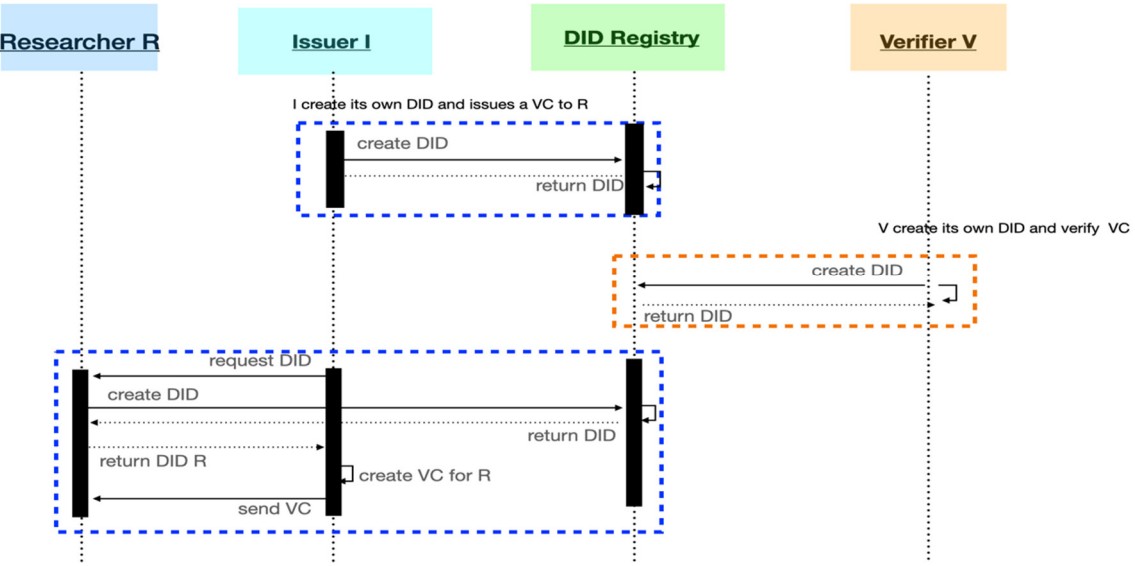

**Figure 7.** Identity and credential of researcher issued by IRB and verified by CDM provider (play a role as verifier).

### 3.5. Prove the Credential

The DID owner transmits a claim that can verify his/her identity as a response to the verification request of the verifier; that is, when the verifier performs verification using the public key of the DID document. Data extracted from one or more credible credential authorized by one or more issuer and exchanged with a particular verifier constitutes a verifiable presentation. For instance, one or more attribute in VC is used to present a proof. Figure 8 shows the process of issuing and verifying a credential in a cloud CDM. The proof contains a presentation (named claim) of information that is given from the credentials that can be verified and are issued from IRB by a body that is respected by the verifiers, the CDM provider, and the CDM consumer. For instance, the name, role, and IRB document number may be presented to the CDM provider as proof. In the proving process, a signature is used to check whether it is a valid VC, and the value of VC is not modified. Hence, proof contains the signature value to verify this verifiable credential. When a verifier receives a verifiable credential, the verifier can verify the signature of the credential to ensure it has been issued by the real issuer and that it has not been tampered with.

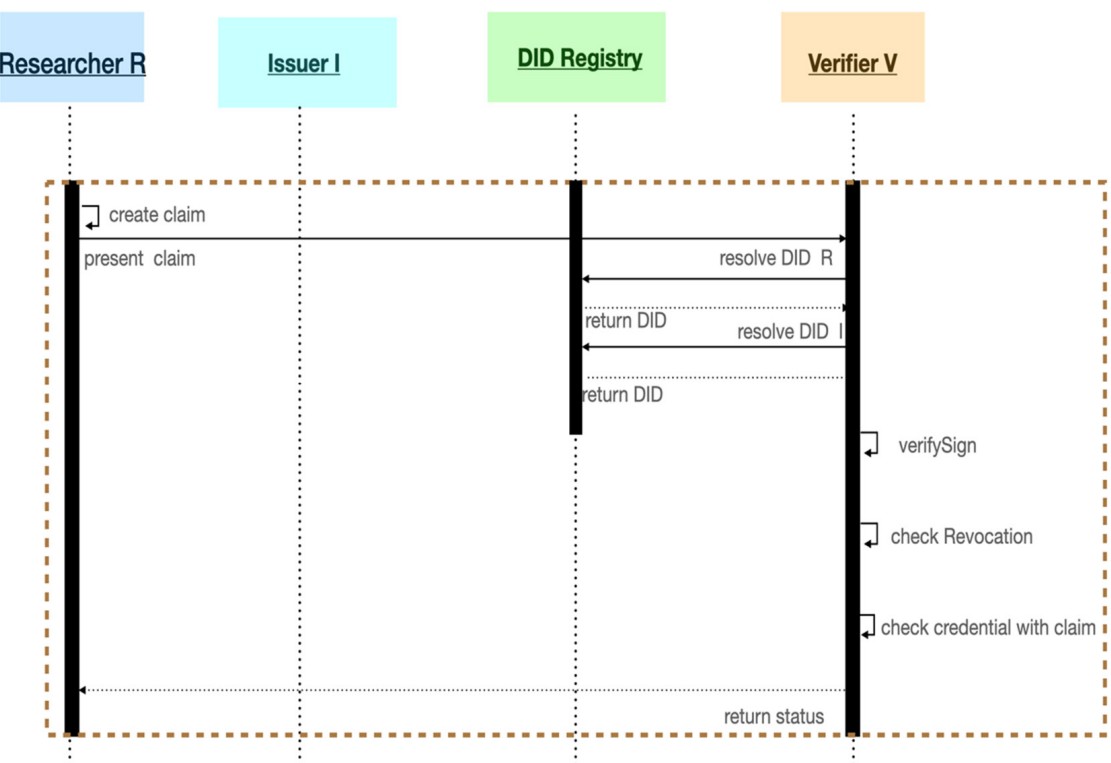

**Figure 8.** Identity of researcher issued by IRB and verified by CDM provider (play a role as verifier).

### 3.6. Discussion

In this sub-section, we analyze a detailed SSI-based scheme of cloud CDM. In order to provide these essential functions of CDM described in Section 3.2, user attributes associated with a DID are preserving privacy in a verifiable credential in a user wallet. As described in Section 3.3, the updating of these attributes and the revoking of DID are operated in the holder. A VC is associated with user attributes. As described in Section 3.4, the privacy sensitive information is handled and stored as the VC in the user's wallet. Hence, there is little risk of a security leak if a researcher keeps this information in his/her wallet. To receive permission to access SP, a verifiable credential is required to exchange exclusively between a prover and a verifier using the process detailed in Section 3.5. To provide a tamper-proof attribute in the proposed scheme, the SP plays a role as a verifier, which verifies the signature of the credential issued by the trustworthy issuer such as IRB by using a decentralized PKI infrastructure.

## 4. Implementation

### 4.1. Experimental Setup

To operate the von-network and agents, a docker engine is managed by containers for running agents. Every container functions as a thin virtual computer. The management of Hyperledger Indy nodes is permissioned [27,28]. Each Hyperledger Indy node is assigned its own ledger, which it uses to store and read publicly available data from the distributed ledger that is consistently chosen. To agree (achieve consensus) on which transactions ought to be written first and in what order, the nodes communicate. A portable version of Hyperledger Indy with a ledger browser termed a von-network is used to launch Hyperledger Indy nodes. The von-network serves as a Hyperledger Indy public ledger sandbox instance. Figure 9 shows the result of consensus which is based on the Redundant Byzantine Fault Tolerance (RBFT) algorithm. The consensus of Indy is used to preserve an agreement on the order of transactions.

```
View_change_status:
  IC_queue: {}
  Last_complete_view_no: 0
  Last_view_change_started_at: '1970-01-01 00:00:00'
  VCDone_queue: {}
  VC_in_progress: false
  View_No: 0
did: 4PS3EDQ3dW1tci1Bp6543CfuuebjFrg36kLAUcskGfaA
verkey: 68yVKe5AeXynD5A8K91aTZFjCQEoKV4hKPtauqjHa9phgitWEGkS5TR
Pool_info:
  Blacklisted_nodes: []
  Quorums: '{''n'': 4, ''f'': 1, ''weak'': Quorum(2), ''strong'': Quorum(3), ''propagate'':
    Quorum(2), ''prepare'': Quorum(2), ''commit'': Quorum(3), ''reply'': Quorum(2),
    ''view_change'': Quorum(3), ''election'': Quorum(3), ''view_change_ack'': Quorum(2),
    ''view_change_done'': Quorum(3), ''same_consistency_proof'': Quorum(2), ''consistency_proof'':
    Quorum(2), ''ledger_status'': Quorum(2), ''ledger_status_last_3PC'': Quorum(2),
    ''checkpoint'': Quorum(2), ''timestamp'': Quorum(2), ''bls_signatures'': Quorum(3),
    ''observer_data'': Quorum(2), ''backup_instance_faulty'': Quorum(2)}'
  Reachable_nodes:
  - - Node1
    - 0
  - - Node2
    - 1
  - - Node3
    - null
  - - Node4
    - null
  Reachable_nodes_count: 4
```

**Figure 9.** The result of consensus of transaction which is stored in their ledgers in nodes.

### 4.2. Experimental Result

Figure 10 shows that a transaction can be considered as the creation of a new DID when a researcher is registered to a DID registry running on the von-network. The transaction gives several pieces of information containing a public key (Verkey) and a DID (Nym) in the distributed ledger. Nym is associated with the legal identity of an identity owner. After registering IRB, a transaction as shown in Figure 11 is generated.

Figure 12 shows an example of a researcher's DID that is stored in his/her wallet. The DID method is used for specifying the mechanism by which a particular type of DID and its associated DID document are created. The DID is a kind of pairwise DID that is intended to be known by its subject and exactly one other party. It is used for operating a secure decentralized authentication mechanism by using asymmetric cryptography technology. In this case, it is resolvable by two parties, the researcher and IRB.

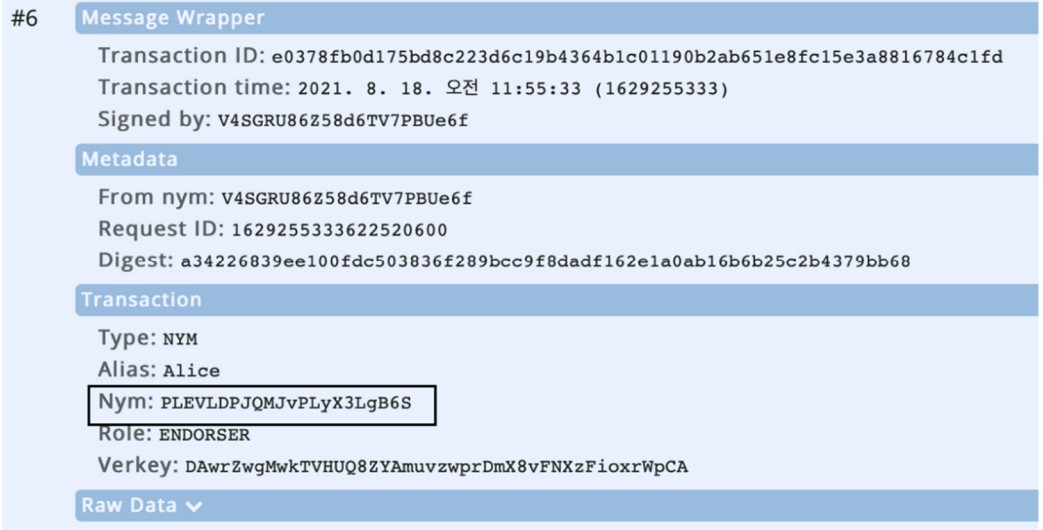

**Figure 10.** The result of a transaction when registering a researcher in von-network.

**#7**

**Message Wrapper**

Transaction ID: 95c066928cfa9bcda4bf61020529477f252240ad40b63661bf077926bf3993f6
Transaction time: 2021. 8. 19. 오전 10:40:28 (1629337228)
Signed by: V4SGRU86Z58d6TV7PBUe6f

**Metadata**

From nym: V4SGRU86Z58d6TV7PBUe6f
Request ID: 1629337228147380200
Digest: 197d2a3a057ca6ae20e053a32d56b74c8ffc3b8fe85c7ce38a039dfc510b39d0

**Transaction**

Type: NYM
Alias: irb.agent
Nym: 24jeM9wbjv7fATku5hrVQi
Role: ENDORSER
Verkey: aeaDNFRB7hcHLgmsJfKYz99tQDQ5esRG1tdMFrp7GJv

**Raw Data** ⌄

**Figure 11.** The result of a transaction when registering IRB in von-network.

```
{
    "results": [
        {
            "did": "WgWxqztrNooG92RXvxSTWv",
            "key_type": "ed25519",
            "method": "sov",
            "posture": "wallet_only",
            "verkey": "H3C2AV…6z3wXmqPV"
        }
    ]
}
```

**Figure 12.** The example of the DID of a researcher.

To issue a credential, the structure of the credential is defined by a credential definition and its schema. The subject of access control such as IRB, CDM provider, and CDM consumer should store them on a von-network. Figure 13 presents an example of an IRB credential schema.

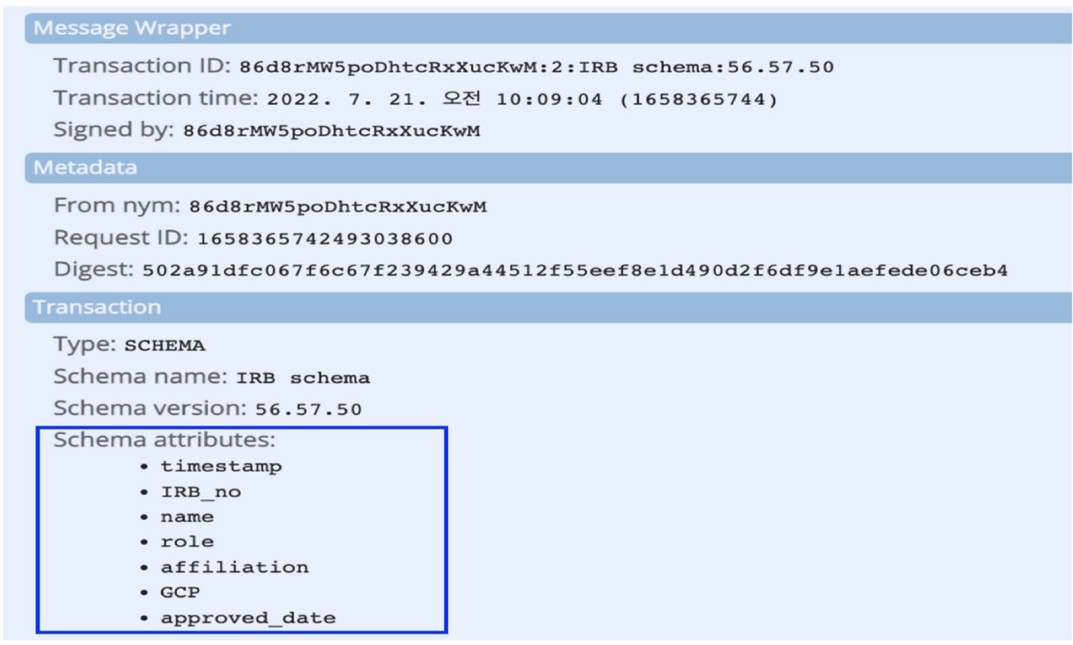

**Figure 13.** The result of a transaction of a credential schema, which is used for issuing a credential by IRB.

To build a web interface for access control of cloud CDM services, flask is used as a web application framework. Hyperledger Aries Cloud Agent Python (ACA-Py) [25] is used as a framework for constructing a VC ecosystem. As shown in Figure 14, the web page illustrates that IRBs give invitation messages to researchers. The message is used to establish a connection between researchers and IRBs. As shown in Figure 15, an IRB should accept a request for invitation by taking the invitation message from a researcher.

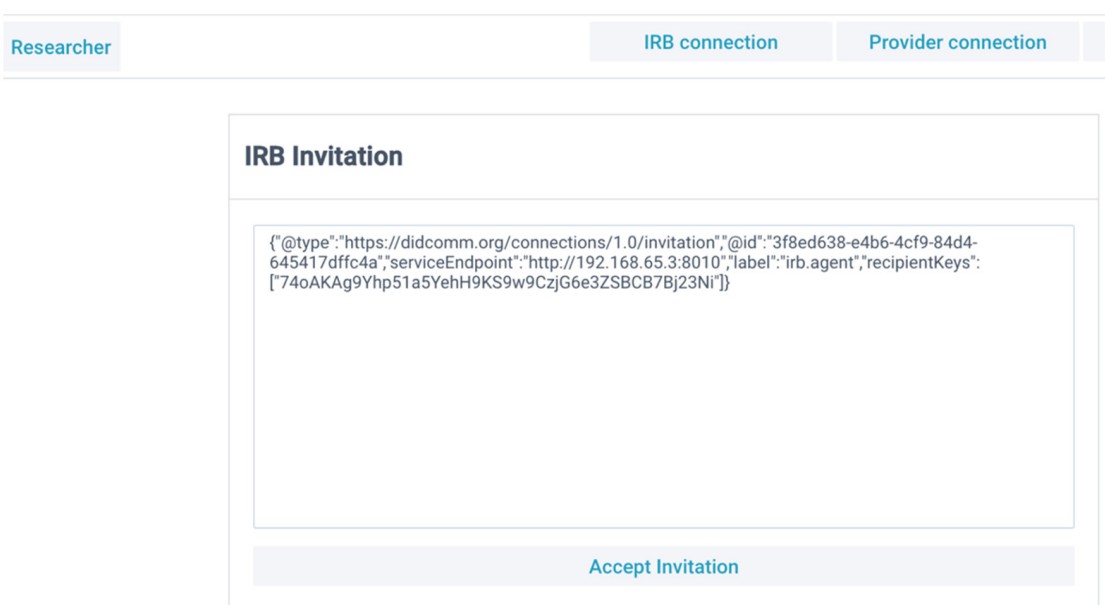

**Figure 14.** The invitation message which is generated by IRB.

In response to the request, the IRB provides a VC with access to the CDM provider. It plays a role as a claim to prove the researcher's identity when connecting to the CDM provider. The researcher obtains permission to make CDM data via a CDM service from a CDM provider. The CDM provider issues a credential to take access of the CDM consumer.

The credential contains two keys and a seed. The seed is used to introduce randomization at the beginning of the CDM data encryption process. At the beginning of the chained block encryption process, the IV introduces randomization. Before encrypting the CDM data, the CDM data is separated into two parts. A symmetric key is used to encrypt each of the two CDM components, ensuring the privacy of the CDM data. Figure 16 shows the VC offered by the CDM provider after delivering encrypted CDM. The VC is used as proof to obtain permission to receive CDM data from the CDM consumer.

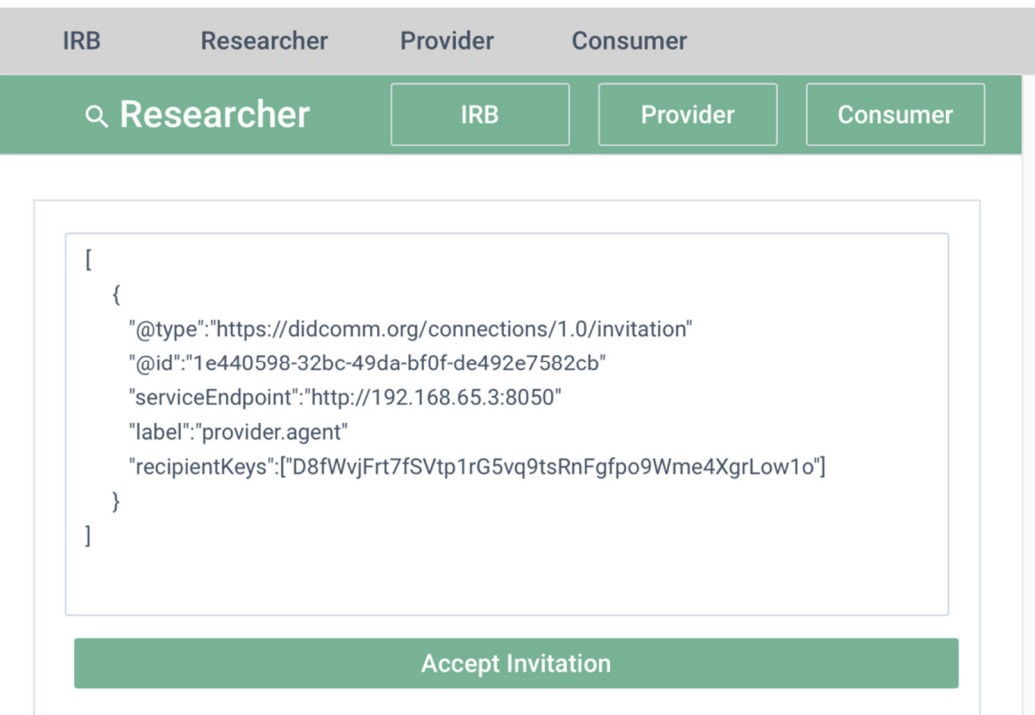

**Figure 15.** Accepting the invitation message by Researcher.

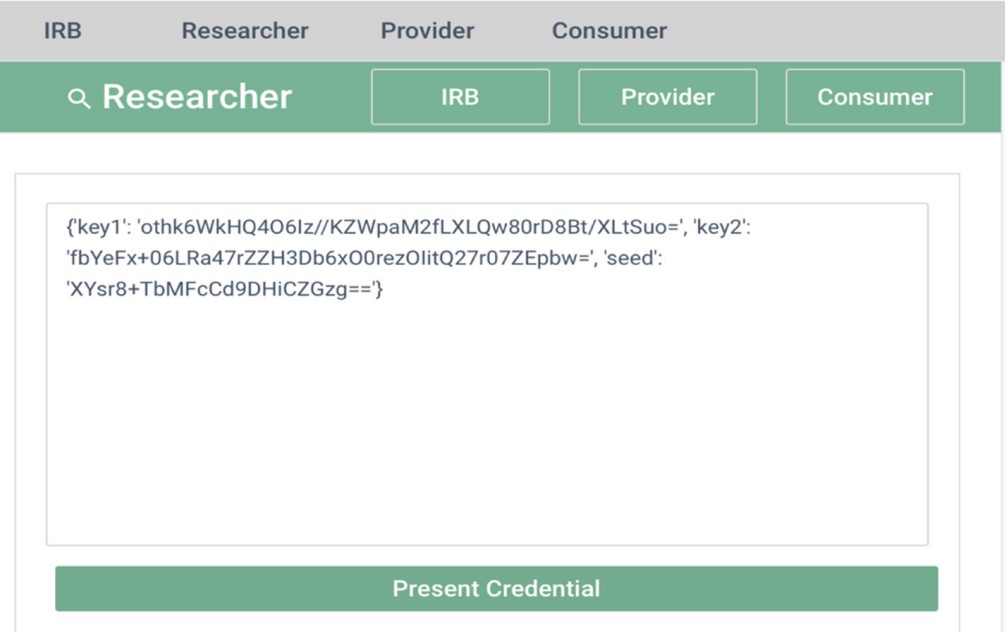

**Figure 16.** VC offered by the CDM provider after delivering CDM encrypted.

As shown in Figure 17, the researcher receives the decrypted CDM data from IPFS when the proof issued by the CDM provider is presented to the CDM consumer. We observe that access to the CDM can be enabled through the user's request, which has been verified through the credentials.

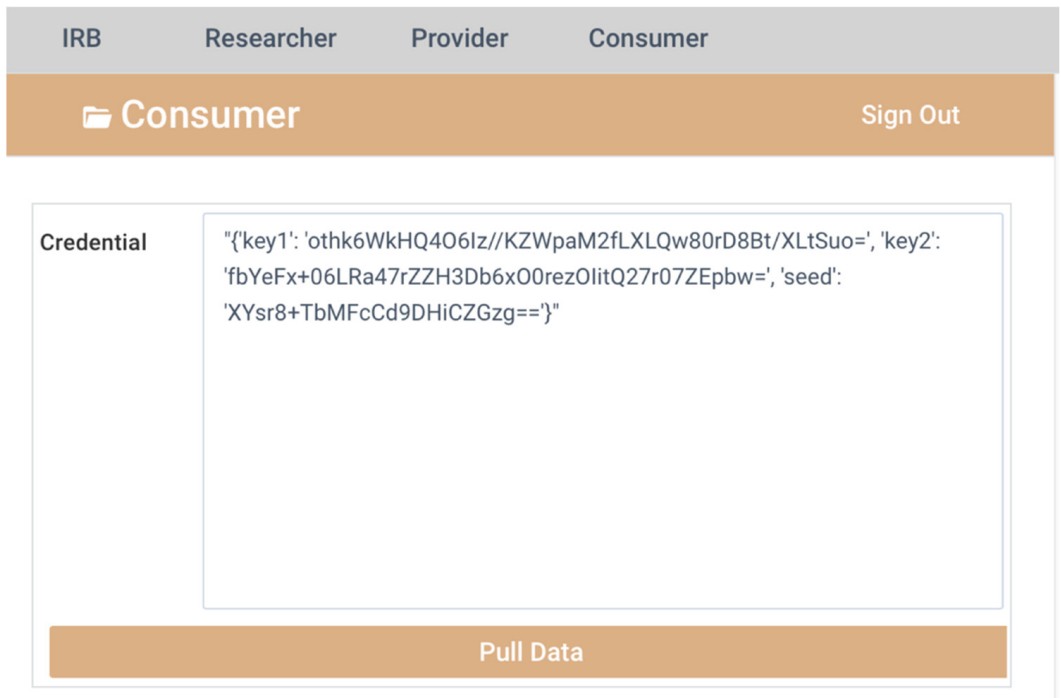

**Figure 17.** The credential which is presented to the CDM consumer as proof.

## 5. Conclusions

Digital identity takes on a significant role as the body of information about an individual in a cloud CDM. Protecting their own data and the privacy of researchers is the main concern of identifying users and their agents for building a cloud CDM. To build a service in a cloud CDM, the access control system is necessary to establish both the interoperability of the systems at the participating institution and the researcher's authorization to take part in the research. To solve this, we designed an access control mechanism by using capability. Capability is associated with an authentication credential that can be verified by the cloud CDM. By asserting claims with the identity and its capability providing its verifiable credential to the authentication entity in the cloud CDM, credibility is established.

The capability based CDM management system is designed and implemented to support well-controlled access control in the cloud CDM. In the case of multiple medical organization data access control, a credential for each researcher is defined for access services related to CDM. In this designed system, credentials are designed as capabilities and presented to access a service given by institutes in the cloud CDM. In the perspective of use cases, we give a partial result of operations of the secure data manipulation based on the CDM management system to support well-controlled access control. It is mainly focused on preserving the security and the integrity of the CDM data.

The implemented system can only be operated in the use case described in the previous section. In the future, the implemented system will be extended to be used in an environment where the subject of identity and the subject of information access are different, such as in the IoT environment.

**Author Contributions:** Conceptualization, Y.B.P. and Y.K.; methodology, Y.B.P.; software, Y.K.; validation, Y.K. and Y.B.P.; formal analysis, Y.K.; investigation, Y.B.P.; resources, Y.B.P.; data curation, Y.B.P.; writing—original draft preparation, Y.K.; writing—review and editing, Y.B.P.; visualization,

Y.K.; supervision, Y.B.P.; project administration, Y.B.P.; funding acquisition, Y.B.P. All authors have read and agreed to the published version of the manuscript.

**Funding:** This research received no external funding.

**Institutional Review Board Statement:** Not applicable.

**Informed Consent Statement:** Not applicable.

**Acknowledgments:** This research was supported by the MSIT (Ministry of Science and ICT), Korea, under the ITRC (Information Technology Research Center) support program (IITP-2022-2017-0-01628) supervised by the IITP (Institute for Information and Communications Technology Promotion).

**Conflicts of Interest:** The authors declare no conflict of interest.

## Abbreviations

The manuscript uses the following abbreviations:

| | |
|---|---|
| CDM | Common Data Model |
| OHDSI | Observational Health Data Sciences and Informatics |
| IPFS | Inter Planetary File System |
| HIS | Hospital Information System |
| DLT | Distributed Ledger Technology |
| GDPR | General Data Protection Regulation |
| RTBF | Right To Be Forgotten |
| IRB | Institutional Review Board |
| SP | Service Provider |
| IdP | Identity Provider |
| IdM | Identity Management |
| ACL | Access Control List |
| SSO | Single Sign On |
| SSI | Self-Sovereign Identity |
| DID | Decentralized Identifiers |
| VC | Verifiable Credential |
| RBFT | Redundant Byzantine Fault Tolerance |
| ACA-Py | Aries Cloud Agent Python |

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
