# Peer review of "Secure Access Control Realization Based on Self-Sovereign Identity for Cloud CDM"

_applsci, doi:10.3390/app12199833_

Round 1

Reviewer 1 Report

Article requires the following suggestions to imporve the readability.

1. Include recent literature in identity management. More literature survey to be added as the existing study is very minimum. this will improve the base for proposed model.

2. comparison of various ealreir proposed models and their conerns to be addressed in a table.

3. Results of the proposed model to be compared with earlier models to convince the reader.

Author Response

  1. Include recent literature in identity management. More literature survey to be added as the existing study is very minimum. this will improve the base for proposed model.

=> To improve the readability, we add recent literature and summarization (including table 1) of the identity models in section 2.

  1. comparison of various earlier proposed models and their concerns to be addressed in a table.

=> We add Table 2 to describe the comparison. Table 2 summarizes the qualitative comparison of the characteristics of the baseline CDM model and the proposed cloud CDM model

  1. Results of the proposed model to be compared with earlier models to convince the reader.

=> We add the Table 2 to describe the comparison models.

Reviewer 2 Report

This paper presents a secure access control scheme using Common Data Model (CDM). The CDM is accessed by the user based on Self-sovereign identity (SSI) where the users shares confidential tokens for identity. I have following observations:

1. Motivation and Contributions in the Introduction Section are not properly written.

2. CDM should be described first before use.

3. It uses Capability-based model, no discussion on capability of a user?

4. The proposed scheme is not compared with the existing scheme.

5. Abbreviations and notations should be written in a Table.

6. Literature survey is missing.

7. Work should be compared with the similar existing work.

8. Security of the scheme should be analysed.

Author Response

  1. Motivation and Contributions in the Introduction Section are not properly written.

=> We correct the subsection named “Motivation and Contributions” in Section 1.

  1. CDM should be described first before use.

=> We add the overview of CDM in section 1.

  1. It uses Capability-based model, no discussion on capability of a user?

=> Since discussion of Capability-based model itself needs a long discussion, we decide it is not proper topic this time. To make clear and eliminate ambiguity in the context of this manuscript, Capability is replaced with SSI in the title of Section 3.

  1. The proposed scheme is not compared with the existing scheme.

=> We add the Table 2 to describe the comparison. Table 2 summarizes the qualitative comparison of the characteristics of the baseline CDM model and the proposed cloud CDM model.

  1. Abbreviations and notations should be written in a Table.

=> We add the abbreviations as a table

  1. Literature survey is missing.

=> we add summarization (including table 1) of the identity models in section 2.

  1. Work should be compared with the similar existing work.

=> We add the Table 2 to describe the comparison. Table 2 summarizes the qualitative comparison of the characteristics of the baseline CDM model and the proposed cloud CDM model

  1. Security of the scheme should be analysed.

- we add a new sub-section 3.6 named discussion. In this sub-section, security of the proposed scheme is described.

Reviewer 3 Report

The paper needs some modification:

1- The abstract should be written part of it is repeated in the introduction.

2- The introduction is not well written.

3- The experimental results are not well explained.

Author Response

The paper needs some modification:

  1. The abstract should be written part of it is repeated in the introduction.

=> We correct and rewrite abstraction

  1. The introduction is not well written.

=> We add the overview of CDM in section 1.

  1. The experimental results are not well explained.

=> In section 4, we describe more details about Figure 12 and Figure 17.

Round 2

Reviewer 2 Report

Updates look satisfactory. 

Reviewer 3 Report

The authors have addressed all the required comments